# Influence of Selective Extraction/Isolation of Heme/Hemoglobin with Hydrophobic Imidazolium Ionic Liquids on the Precision and Accuracy of Cotinine ELISA Test

**DOI:** 10.3390/ijms232213692

**Published:** 2022-11-08

**Authors:** Jolanta Flieger, Małgorzata Tatarczak-Michalewska, Wojciech Flieger, Jacek Baj, Grzegorz Buszewicz, Grzegorz Teresiński, Ryszard Maciejewski, Jacek Wawrzykowski, Dominika Przygodzka, Valery Lutsyk, Wojciech Płaziński

**Affiliations:** 1Department of Analytical Chemistry, Medical University of Lublin, Chodźki 4A, 20-093 Lublin, Poland; 2Department of Anatomy, Medical University of Lublin, Jaczewskiego 4, 20-090 Lublin, Poland; 3Department of Forensic Medicine, Medical University of Lublin, Jaczewskiego 8B, 20-090 Lublin, Poland; 4Department of Biochemistry, Faculty of Veterinary Medicine, University of Life Sciences in Lublin, Akademicka 12, 20-033 Lublin, Poland; 5Jerzy Haber Institute of Catalysis and Surface Chemistry, Polish Academy of Sciences, Niezapominajek 8, 30-239 Cracow, Poland; 6Department of Biopharmacy, Medical University of Lublin, Chodźki 4a, 20-093 Lublin, Poland

**Keywords:** imidazolium ionic liquids, hemoglobin, hem, extraction, cotinine, serum

## Abstract

In this study, ionic liquids were used for the selective extraction/isolation of hemoglobin from human serum for cotinine determination using the ELISA Kit. The suitability of hydrophobic imidazolium-based ionic liquids was tested, of which OMIM BF_4_ (1-methyl-3-octylimidazolium tetrafluoroborate) turned out to be the most suitable for direct extraction of hemoglobin into an ionic liquid without the use of any additional reagent at one extraction step. Hemoglobin was separated quantitatively (95% recovery) from the remaining types of proteins remaining in the aqueous phase. Quantum mechanical calculations showed that the interaction of the iron atom in the heme group and the nitrogen atom of the ionic liquid cation is responsible for the transfer of hemoglobin whereas molecular dynamics simulations demonstrated that the non-covalent interactions between heme and solvent are more favorable in the case of OMIM BF_4_ in comparison to water. The opposite trend was found for cotinine. Selective isolation of the heme/hemoglobin improved the ELISA test’s accuracy, depending on the cotinine level, from 15% to 30%.

## 1. Introduction

Despite increasing social awareness, tobacco smoking is still one of the main causes of mortality and morbidity from cardiovascular diseases, lung diseases, and cancer [1,2]. Recent investigations have shown other results of cigarette smoking in the form of increased risk of postoperative complications, including infection, impaired wound healing, recovery time, and cardiovascular complications [3]. An accurate assessment of tobacco smoke exposure or the effects of environmental smoke is crucial for documenting the patient’s condition and assessing the risk of smoking-related diseases [4]. A standardized questionnaire (survey) is usually used for this purpose; however, it is not always reliable due to the bias of the respondents. For this reason, the determination of biological markers is required to rigorously evaluate exposure status [5]. There are a number of chemicals, that arise from known metabolic pathways of nicotine, but six of them belong to primary ones, namely: 3′-hydroxycotinine, 5′-hydroxycotinine, cotinine *N*-oxide, cotinine methonium ion, cotinine glucuronide, and norcotinine [6]. Quantitatively, the most important metabolites of nicotine in mammalians are cotinine and trans-3-hydroxycotinine, present in blood plasma, urine, and saliva.

The half-life of nicotine is approximately 2–3 h, whereas that of cotinine is 12–20 h [7,8,9,10]. This is the reason why cotinine, due to the longer half-life, is recommended by the Society for Research on Nicotine and Tobacco (SRNT), as the main biomarker of tobacco use (besides carbon monoxide and thiocyanates) in clinical trials [11]. Cotinine levels in serum, saliva, or urine are the basis for distinguishing smokers from non-smokers. Setting a limit value still remains a problem. In 2016, Kim [12] collected epidemiological data from 1985–2014 devoted to the level of this biomarker. The prepared compilation made it possible to distinguish the range of concentrations. Based on these data, the cut-off value for serum cotinine concentration was established in the range of 10–20 ng mL^−1^.

The most used methods of quantifying nicotine and its metabolites are gas (GC) and liquid (LC) chromatography [13,14,15,16,17,18]. It should be emphasized, however, that these methods require a tedious procedure of sample preparation to avoid disadvantaged matrix effects. Moreover, in order to obtain an LOD on the level of ng/mL, additional derivatization steps [19] or sensitive detection involving flame ionization (FID), or mass spectrometry (MS) should be used. In this way, cotinine can be determined at the level of 2 ng mL^−1^ [16], 0.1 ng mL^−1^ [17], and even at the level of 0.050 ng mL^−1^ using the UHPLC-QQQ-MS/MS method [18]. In addition to the above-mentioned advanced separation techniques, immunological tests with high specificity, e.g., radioimmunoassay (RIA), enzyme immunoassay (ELISA), and immunofluorescence test (FIA) are used to quantify cotinine in biological fluids [20,21,22,23,24,25,26,27,28,29,30,31]. For example, a quick, accurate, and cheap “Cotinine ELISA Kit” has a sensitivity of 1 ng mL^−1^ in the measured range of 0–100 ng mL^−1^ and allows the determination of cotinine in serum, urine, and saliva [32].

It should be emphasized that in the case of serum, samples showing evidence of hemolysis are rejected. According to the analytical and clinical decision limits for hemolysis, the reporting range for hemoglobin should be less than 1 mg mL^−1^. In this case only, there is no need to comment and confirm the obtained results [33]. The presence of greater amounts of hemoglobin causes spectroscopic interference, which disturbs absorption measurements in biochemical analyzes, resulting in incorrect results.

Hemolyzed samples are problematic and not recommended also for immunological tests, especially those designed for measurements of unstable analytes such as insulin, glucagon, calcitonin, parathyroid hormone, ACTH, and gastrin due to the release of proteolytic enzymes that degrade them. Hemolysis can also interfere with some steps in signal generation in immunoassays. Free hemoglobin can bind nonspecifically to the solid phase resulting in a false positive. In 2021, Wang et al. [34] confirmed the effect of hemolysis on the analysis of total IgE with a commercial ECLIA immunoassay. Elimination of hemoglobin from lysates (blood or erythrocytes) is necessary for proteomics by mass spectrometry or electrophoresis [35], as it interferes with the quantification of other proteins. Therefore, it is a good analytical practice to avoid serum samples with traces of hemolysis as the effect of this on the test result is never certain [36,37,38].

Hemolysis is the rupture of the red blood cell (RBC) membrane, releasing hemoglobin and other internal components into the surrounding fluid. Hemolysis is a common phenomenon that gives the serum or plasma a pink to red hue. The process of hemolysis may take place in vivo as a result of autoimmune hemolytic anemia, as a post-transfusion reaction [39], and in vitro as a result of improper collection, handling, or transport of samples [40]. In recent years, it has been shown that free heme, the prosthetic group of hemoglobin, can also accumulate in erythrocytes and is released during the lysis of red blood cells (RBC) [41]. The mean concentration of free heme in erythrocytes was estimated at 21 ± 2 μM. It should be noted that the release of heme depends on the hemoglobin concentration [42]. At low hemoglobin concentrations, its tetrameric forms degrade into dimers, and heme is released more quickly [43]. Thus, heme present in the hemolysate is not only derived from the cytosol of red blood cells but can be released upon dilution of the hemoglobin. Unbound with heme protein, it exhibits a cytotoxic effect due to iron redox activity and the hydrophobicity of protoporphyrin [44,45].

The selective removal of heme/hemoglobin traces from serum is an analytical challenge and may have an influence on the reliability of various analytical tests. BioTech Support Group, (Deer Park Drive, Suite M, Monmouth Junction, NJ, USA) offers HemogloBind for this purpose, which is a biopolymer in gel and suspension form, which is specially designed to remove hemoglobin from serum and plasma samples. Highly specific removal of hemoglobin can be achieved by immunological methods [46] or by selective precipitation with organic solvents, e.g., 8% (*v*/*v*) 1-butanol in ethanol or zinc chloride in a 10–15 fold molar excess to hemoglobin [47]. Precipitation methods, however, change the composition of the serum.

To separate histidine-rich proteins, such as hemoglobin, affinity chromatography on transition metal ions (Cu^2+^, Ni^2+^, Zn^2+^, or Co^2+^) immobilized on matrices can be applied [48]. Metals could also be immobilized on the surface of magnetic nanoparticles (MNP) [49]. Magnetic cores, usually made of iron oxides, are coated with SiO_2_ for protection against acidic environments [50]. An example of such a composite is CoFe_2_O_4_, [51]. More recently hierarchical copper shells anchored on magnetic nanoparticles were designed to selectively deplete hemoglobin from human blood [52]. The batch experiments were conducted using bovine hemoglobin (BHb). The adsorption increased linearly with increasing initial protein concentration up to 1.0 mg mL^−1^. The adsorption capacity was estimated to be 666 mg BHb per 1 g of Cu-MNPs. The system however required a long equilibration time of about 90 min and 100-fold sample dilution in 0.1 M Tris-HCl pH 8.5.

Another possibility of the purification and separation of proteins offers extraction systems based on ionic liquids such as liquid–liquid extraction utilizing hydrophobic ionic liquids, two-phase aqueous systems (ABS) consisting of hydrophilic ionic liquids, and supported ionic liquids (SILs), in which ILs are modifying different solid materials as covalently bonded ligands. Excellent review articles [53,54] have been provided on this subject. In 2008, Cheng et al. [55] for the first time extracted hemoglobin from serum samples by a new synthesized ionic liquid 1-butyl-3-trimethylsilyl-imidazolium hexafluorophosphate (BtmsimPF_6_). The authors managed to remove 100 ng µL^−1^ of hemoglobin with an efficiency of 93%. However, the preparation of BtmsimPF6 required a tedious, time-consuming, procedure with the use of different organic solvents such as chlorobutane, toluene, ethyl acetate, and controlling the temperature. The yield at the last stage of the multi-stage synthesis process, namely, the conversion of chloride to hexafluorophosphate ionic liquid, achieved 85%.

Despite the existence of a few techniques for removing hemoglobin from serum, there is still a need for simple, cheap, readily available, ecological methods that reduce the use of organic solvents and ensure satisfactory efficiency. The aim of the work is to obtain good-quality serum samples for the determination of cotinine concentration using the highly specific ELISA test by removing trace amounts of heme and/or hemoglobin. For this purpose, different commercially available imidazolium ionic liquids were tested. Reliable determination of the cotinine content especially at a lower concentration level is of particular importance in distinguishing between smokers and non-smokers.

Additionally, a series of molecular modeling methods (quantum mechanical calculations, classical molecular dynamics simulations, and thermodynamic integration calculations) was applied to provide a molecular interpretation of processes underlying the investigated isolation technique.

## 2. Results and Discussion

### 2.1. Optimization of Extraction Conditions (IL Kind, IL Amount, Extraction Time)

Heme/hemoglobin was extracted from the plasma sample using different ionic liquid-based biphase systems. Each system consisted of two coexisting immiscible aqueous-rich phases. The ionic liquid phase was located in the lower phase while the upper phase was constituted of plasma. Such two-phase systems had already been recognized as an efficient pre-treatment strategy for the extraction of different analytes from aqueous media. The five ionic liquids have been investigated for the extraction of heme/hemoglobin. They have PF_6_ or BF_4_ anions and different cationic imidazolium moieties exhibited hydrophobic properties. Figure 1 presents the relationship of absorbance of the upper aqueous phase versus wavelength after extraction utilizing different ionic liquids such as HMIM BF4: 1-hexyl-3-methylimidazolium BF4; BMIM PF6: 1-butyl-3-methylimidazolium PF6; diMIM PF6: 1,3-dimethoxyimidazolium PF6; EMIM PF6: 1-ethyl-3-methylimidazolium PF6. As the hemoglobin is transferred from the upper aqueous phase into the lower phase after IL addition that is why the absorbance of the upper phase decreases at the wavelength of 410 nm attributed to the presence of hemoglobin in comparison to absorption recorded before extraction. The extraction system was composed of 3 mL of serum containing hemoglobin at a concentration of 100 ng mL^−1^ and 0.5 g of IL. The measurements were performed after 30 min of shaking at room temperature.

The disappearance of the absorption band at 410 nm was achieved by using the most hydrophobic ionic liquid OMIM BF_4_. In the case of remaining ionic liquids, the extraction of heme/hemoglobin was less favorable.

Figure 2 shows the dependence of the absorbance of the serum sample containing hemoglobin on the mass of the OMIM BF_4_ ionic liquid used for the extraction. The best effect of heme and/or hemoglobin isolation is ensured by the addition of an ionic liquid in the range from 0.7 to 1.0 g for no more than 3 mL of serum containing 0.15 mg mL^−1^ of hemoglobin. The above composition of the mixture removed at least 94.89% of hemoglobin. By adding a lesser amount of IL (from 0.1 g to 0.6 g), the extraction efficiency decreases to 54.1% to 88.66% of hemoglobin.

The shaking time of the extraction mixture was optimized. A two-phase system was constructed by mixing 0.7 g of OMIMBF_4_, and 3 mL of serum containing 0.125 mg mL^−1^ hemoglobin. The samples were shaken from 0 to 20 min. After centrifugation, the aqueous upper phase was analyzed spectrophotometrically at 410 nm. The results of the experiment are presented in Figure 3. They show that the time needed for heme/hemoglobin extraction under the conditions of analysis cannot be shorter than 5 min. The kinetics of the process corresponds to the assumed, first-order reversible kinetics, represented mathematically by Equation (1). This model assumes that the rate of solute influx from any liquid phase is proportional to the solute concentration in this phase; such assumption is typical of liquid–liquid extraction. This kinetic model for infinite times (i.e., for the equilibrium state) reduces to linear relationship between solute concentrations in both phases.

### 2.2. The Partitioning of the Heme/Hemoglobin in IL-Based Extraction System

The effect of serum hemoglobin concentration on the extraction efficiency in the tested system (3 mL of serum + 0.7 g IL) is shown in Figure 4. As can be seen, the hemoglobin content of 0.125 mg mL^−1^ is removed with 94.89% efficiency. Further increases in hemoglobin concentration deteriorates the efficiency of extraction.

The simplest model of liquid–liquid extraction, assuming linear relationship between equilibrium concentrations of solute in both liquid phases (equivalent to constant separation ratio) works reasonably well only for lower concentrations of solute (note that the upper limit is higher that the equilibrium concentration resulting from kinetic experiments, Figure 3). This linear dependence is disturbed upon the increase in solute concentration which may be interpreted (in qualitative terms) as the result of some saturation effects that occur in ionic liquid phase. Despite several attempts, we were unable to propose the uniform, equilibrium model that would: (i) work within the whole range of concentrations; (ii) have a well-defined physical meaning; (ii) provide a set of best-fitted parameters of realistic values. Therefore, the purely empirical, hyperbolic relationship between separation factor and solute concentration in water was used instead for higher concentration. Figure 4 illustrates how the two mathematical models work in relation to the experimental data.

### 2.3. Serum and Hemoglobin Enriched Serum Electrophoresis

Selective removal of hemoglobin by means of an ionic liquid was confirmed by SDS-PAGE electrophoresis. Electrophoretic profiles obtained for serum enriched with hemoglobin at a concentration of 0.15 mg mL^−1^ before and after extraction with the OMIM BF_4_ are illustrated in Figure 5A,B. The separated bands were compared to a standard protein mixture that resolves depending on molecular weight in the range of 10–250 kDa when electrophoresed (Figure 5C). Since albumin accounts for two-thirds of the plasma proteins, electrophoresis was performed after removal of albumin (Figure 5D,E) and twenty-fold concentration (Figure 5F,G) to show the selectivity of the extraction with respect to hemoglobin. In serum samples before (D) and after (E) extraction with OMIM BF_4_, no differences in band intensity were observed. On the other hand, the hemoglobin standard (Figure 5F) shows a significant reduction in the intensity of the bands in the range of 64–68 kDa after extraction (Figure 5G). It clearly confirms the selective removal of hemoglobin by the ionic liquid.

### 2.4. Molecular Modeling of Interactions between IL and Heme/Hemoglobin/Cotinine

#### 2.4.1. Quantum Mechanical Calculations

The quantum mechanical calculations allow to identify possible heme–EMIM complexes, created by the formation of a relatively weak coordination bond involving an iron cation of heme and nitrogen atoms of EMIM. Not including the possible conformational heterogeneity of final complexes, we distinguished only one possible complexation pattern, namely, the 1:1 complexation involving N_3_ nitrogen atom of EMIM (i.e., the nitrogen atom attached to the methyl moiety). The analogous hypothetical bond involving N_1_ instead of N_3_ is unstable and leads to non-covalent interactions of heme and EMIM. Interestingly, the complexes of the 2:1 stoichiometry, involving the two separate EMIM cations were also impossible to obtain and usually resulted in either obtaining complexes of the 1:1 type or non-covalent associates of EMIM and heme. Both the geometry of the formed complex (the Fe-N_3_ distance equal to 0.215 nm, comparable to 0.2 nm for analogous Fe-N pairs in heme) and the associated energy change (−77.38 kJ/mol) suggest the formation of a coordination bond. Although the energy change is relatively small, its magnitude is sufficient to confirm the favorability of the complexation process. Moreover, the order of tens of kJ/mol is typical for energies associated with the Fe-N bond in organic complexes [56]. The optimized structure of the heme–OMIM complex is presented on Figure 6.

#### 2.4.2. Molecular Dynamics Simulations

The purpose of thermodynamic integration procedure was to determine the favorability of the interactions of the two solutes (heme and cotinine) with two solvents (water and OMIM+BF_4_). By subtracting the free energy changes computed for corresponding fragments of the thermodynamic cycle (see the Methods section for details) one can determine the free energy change associated with the transfer of a solute molecule from one solvent to another. The following values were determined for the process of transfer of solute molecule from water to OMIM+BF_4_ for heme and cotinine, respectively: −127.3 ± 5.1 kJ/mol and 5.0 ± 1.1 kJ/mol. The former value confirms that, independently of the results of quantum mechanical calculations, additional factors favoring the transfer of heme from water to ionic liquid are the non-covalent interactions. On the contrary, the latter, cotinine-related value, having the opposite sign, confirms that this compound prefers the aqueous environment over the ionic liquid. Such opposite trends are in line with the results of the experimental study and explain the mechanism behind the process of heme isolation even without evoking the hypothesis about the formation of heme–ionic liquid cation coordination bonds.

The analogous, thermodynamic integration-based procedure was carried out for the whole hemoglobin molecule immersed in either water or OMIM+BF_4_. The free energy change associated with the process of transfer of hemoglobin molecule from water to OMIM+BF_4_ is equal to −38.6 ± 15.1 kJ/mol. Keeping in mind that in this case the thermodynamic integration is probably less accurate than for much smaller molecules of heme and cotinine (due to large size of hemoglobin molecule, associated less intensive sampling and artifacts related to volume changes), one has to note that the observed free energy change (and its sign) speaks to larger affinity of hemoglobin to IL in comparison to water.

Additionally, we compared the influence of the presence of an alkyl chain in the ionic liquid cation on the favorability of interactions between such cation and the heme molecule, under assumption that (according to the results of the quantum mechanical calculations) the coordination bond between N_3_ and Fe is created. Such comparison is reasonable in the context of diverse results obtained for ionic liquids containing either OMIM or EMIM cations (less favorable extraction when using EMIM-containing IL). This stage of molecular modeling study allowed to determine the energy contribution resulting from the presence of elongated alkyl chain to the overall heme–cation interaction energy. This contribution is, on average, equal to ca. −6.9 kJ/mol (calculated as a sum of Lennard–Jones interactions equal to ca. −7.1 kJ/mol and Coulombic interactions are equal to ca. 0.2 kJ/mol). The negative value speaks to more favorable interactions of heme–OMIM in comparison to the heme–EMIM ones. Although the magnitude of the determined energy does not seem to be especially high, it certainly is an additional factor supporting the stability of the heme–OMIM associates.

### 2.5. Cotinine ELISA Kit

To perform the “Cotinine Elisa Kit” test, serum samples (3 mL) containing hemoglobin <0.5 mg mL^−1^ (0.05 mg mL^−1^; 0.1 mg mL^−1^; 0.2 mg mL^−1^), which corresponds to the hemolysis index level 1, were used. The serum was supplemented with cotinine and analyzed by ELISA before and after extraction with OMIM BF_4_ (0.7 g). Following the test protocol, the quantification was based on a calibration curve that was prepared for the following cotinine concentrations: 0; 5; 10; 25; 50; 100 ng mL^−1^ (Figure 7). The concentration of 20 ng mL^−1^, lying in the middle of the curve, was chosen to investigate the effect of heme/hemoglobin on the accuracy and precision of the serum cotinine determination. This is a critical value used to distinguish smokers from non-smokers. Moreover, the shape of the calibration curve indicates that the determinations made for the extreme values would be too error-prone.

The results of the measurements together with the statistical parameters (CV%, SD, mean) are presented in Table 1. Based on the obtained results, it can be concluded that the relative error of cotinine concentration measurements improved after removing the hemoglobin present at the level of 0.05 mg mL^−1^ from 16.23% to 10.66 %, i.e., 5.57%; and for the higher hemoglobin content (0.1 mg mL^−1^) it did not change (0.2 mg mL^−1^) or it decreased from 32.18% to −0.09%, i.e., by 32.27%. Overall, the precision and accuracy (CV%, SD) of the individual serum cotinine ELISA measurements improved after the removal of heme/hemoglobin traces.

## 3. Materials and Methods

### 3.1. Apparatus and Reagent

Investigated ionic liquids, namely, 1-Methyl-3-octyl-imidazolium-tetrafluoroborate (OMIM BF_4_), were obtained from Sigma (St. Louis, MO, USA) except for 1-ethyl-3-methyl-imidazolium hexafluorophosphate (EMIM PF_6_), which was from Fluka (Sigma-Aldrch Group, Switzerland). Distilled, deionized water was obtained from Barnstead Deionising System (Dubuque, IA, USA) to be used for preparation of all the solutions. Serum blood and hemoglobin were purchased from Sigma-Aldrich, (St. Louis, MO, USA). Spectrophotometric measurements were made using a GENESYS 20 spectrophotometer (Thermo Fisher Scientific Inc., Waltham, MA, USA).

Cotinine ELISA Kit (Catalog Number KA0930, 96 assays, Version: 08) was purchased from Abnova (Taoyuan City, Taiwan).

### 3.2. The Measurement of Cotinine in Human Serum

A solid phase competitive ELISA Kit was used for the measurement of cotinine in human serum. The measurements were in agreement with the Cotinine ELISA Kit assay procedure. Absorbance was measured at 450 nm by the use of SpectraMax Multi-Mode Microplate Detection System (Molecular Devices LLC. Sunnyvale, CA, USA). The absorbance at 450 nm was inversely proportional to the concentration of cotinine in the samples. The limit of detection was 1 ng mL^−1^. The quantitative determination was based on a standard curve prepared in the range from 5 to 100 ng mL^−1^.

### 3.3. General Procedure of Heme/Hemoglobin Extraction

The ionic liquid was added in the following amounts: 0.1; 0.3; 0.5; 0.6; 0.7; 0.8; 0.9; 1.0 g to 3 mL of serum containing 0.15 mg mL^−1^ (absorbance at 410 nm was approximately 1). The mixture was shaken in the rotator for 15 min at maximum speed (30 RPM), and then centrifuged for 5 min. After complete phase separation was performed, the absorbance of the upper aqueous phase was measured at 410 nm versus reagent blank. The absorbance spectra were recorded in the range from 350 nm to 500 nm. The experiment was repeated three times and results are expressed as mean ± standard deviation. Heme/hemoglobin was extracted into the lower IL-rich phase (Figure 8). The analytical procedure was described in the Polish Patents ZD-17/2022/CTW.

### 3.4. Electrophoresis

SDS-PAGE electrophoresis under denaturing conditions was performed for selected samples. Albumin was removed from the serum using the Pierce ™ Albumin Depletion Kit (85160, (Thermo Fisher Scientific Inc., Waltham, MA, USA).

Hemoglobin standard samples (before and after extraction with the OMIM BF_4_) were concentrated 20-fold using an Ultra 0.5 Centrifugal Filter—Amicon Centrifugal Filter Unit with a 3 kDa cut-off (Merck Millipore, UFC5003, 3 kDa MWCO). The stained gel images were digitized using a ImageScanner III (GE Healthcare Life Sciences). The standard of molecular weights (Bio-Rad, Precision Plus Protein ™t Standards, 10–250 kDa) was used for the bands. Electrophoresis was performed on polyacrylamide gel in Laemmli system [57], gel percentage 12%, voltage 100V (Mini-PROTEAN^®^ Tetra Cell, PowerPac™ HC, Bio-Rad). After electrophoresis, the proteins were stained with silver salts according to the procedure of Shevchenko [58].

### 3.5. Molecular Modeling

#### 3.5.1. Quantum Mechanical Calculations

Quantum mechanical calculations concerned the following types of structures: (1) isolated heme molecule; (2) isolated cation of ionic liquid, 1-methyl-3-ethyl-imidazolium (EMIM), being a simplified representation of the OMIM cation, lacking a long, alkyl chain; (3) a series of various complexes composed of a single heme molecule and 1-2 EMIM cations. The initial structure of heme was taken from PDB database, whereas the structure of EMIM was drawn manually and preoptimized within the UFF force field [59]. EMIM-heme complexes were prepared by using the Avogadro 1.1.1. software [60] by testing various complexation schemes (e.g., 1 vs. 2 EMIM cations per 1 heme molecule) or coordination bonds. Such approach aimed to test whether EMIM cations are capable of creating stable, attractive interactions with an iron ion in heme and determined the magnitude of their strength. All quantum mechanics calculations were performed at the B3LYP/6-311G(d,p) level of theory [61,62,63] by using Gaussian09 software [64] and the energy change associated with the formation of a given heme–EMIM complex was calculated from the simple energy balance equation, neglecting the influence of BSSE.

#### 3.5.2. Molecular Dynamics Simulations

All MD simulations were carried out with the GROMACS 2016.4 package [65] and within CHARMM36 force field [66]. The molecules (cotinine, heme, or hemoglobine) were placed in cubic simulation boxes that were dimension dependent on the system type (box edge varying from ca. 5.9 to ca. 12 nm) and surrounded by the number of explicit solvent molecules accounting for the system density of approximately 1 g/cm^3^ (water) or 1.12 g/cm^3^ (ionic liquid). The structure of the hemoglobin was based on the PDB:1N45 entry in the PDB database. The GROMACS-readable topologies for heme, hemoglobin, ionic liquid, and cotinine were prepared by using the CHARMM-GUI server [67]. The MD simulations were carried out under periodic boundary conditions and in the isothermal–isobaric ensemble. The temperature was maintained close to its reference value (298 K) by applying the V-rescale thermostat [68], whereas for the constant pressure, (1 bar, isotropic coordinate scaling) the Parrinello–Rahman barostat [69] was used with a relaxation time of 0.4 ps. The equations of motion were integrated with a time step of 2 fs using the leap-frog scheme [70]. The hydrogen-containing solute bond lengths were constrained by application of the LINCS procedure with a relative geometric tolerance of 10^−4^ [71]. The electrostatic interactions were modeled by using the particle-mesh Ewald method [72] with cut-off set to 1.2 nm, while van der Waals interactions (LJ potentials) were switched off between 1.0 and 1.2 nm. TIP3P model [73] was applied to represent explicit water.

The Gibbs free energy associated with the transfer of a solute molecule from aqueous to ionic liquid phase was calculated by using the thermodynamic integration (TI) approach [74]. To annihilate the solute from either type of solvent all nonbonded interactions involving solute atoms were scaled down to zero in a stepwise manner as a function of a coupling parameter λ. The associated free energy changes were calculated with the Bennett Acceptance Ratio (BAR) method [75], implemented in the GROMACS gmx bar subroutine, including the error estimation determined by using the default criteria. The 21 evenly spaced λ-points were accepted and the data from equilibrated systems were collected every 0.02 ps for a duration of 10 ns in each λ window. The Coulomb and van der Waals parameters were perturbed simultaneously and a soft-core function was used for the van der Waals interactions to prevent energy singularities. The final value of the free energy change was calculated as the difference between the free energy change calculated for decoupling solute from aqueous solution and that obtained for decoupling the same molecule from ionic liquid.

Additionally, a regular molecular dynamics simulation was performed for the heme+ionic liquid system with a harmonic potential introduced to keep the small (ca. 0.215 nm) distance between N_3_ nitrogen atom of the OMIM cation and iron ion in heme. During the simulation the data concerning non-bonded interactions between alkyl chain of OMIM and associated heme molecule were monitored. The simulation lasted 10 ms and the data were collected every 0.5 ps.

### 3.6. Kinetics and Equilibrium of Extraction

The kinetics of heme extraction was modeled by using the following first-order reversible kinetic equation:(1)$$\frac{dC_{aq}}{dt}=-k_{1}C_{aq}+k_{2}\left(q-C_{aq}\frac{V_{aq}}{V_{IL}}\right),$$
where *C_aq_* is the concentration of heme in aqueous phase, *q* is the total amount of heme present in the system, *V_aq_* and *V_IL_* are volumes of the two phases participating in the extraction process (aqueous and ionic liquid, respectively) whereas *k*_1_ and *k_2_* are rate constants. Except of *k_1_* and *k_2_*, which are unknown, adjustable parameters, the values of remaining variables in Equation (1) are known from operational conditions. Equation (1) was solved for the following boundary condition: *C_aq_*(*t* = 0) = *q*/*V_aq_*. The Mathematica 12 (Wolfram Research) program was used to solve the differential Equation (1) and adjust the best-fit coefficients.

The equilibrium of extraction was modeled in terms of constant ratio between concentrations of heme in both solvent phases (called the separation ratio). It is equivalent to the linear relationship between equilibrium values of *C_aq_* and *C_IL_*. It was found that such relationship does not allow to describe the experimental data in the whole range of investigated concentrations. Therefore, for more concentrated solutions, the purely empirical relationship, of hyperbolic mathematical form was introduced to model the relationship between *C_aq_* and *C_IL_*. It is assumed to reflect the saturation-related processes resulting in the limited solubility of heme in ionic liquid. The results of the mathematical modeling are given in Figure 4.

## 4. Conclusions

The selective removal of hemoglobin from blood lysates may have applications in many fields. The presence of hemoglobin causes spectroscopic interference, which disturbs absorption measurements in biochemical analyzes, and even immunological measurements, resulting in incorrect results.

The paper proves that 1-methyl-3-octylimidazolium tetrafluoroborate (OMIM BF4) is able to extract hemoglobin and/or heme from the serum samples in an efficient manner. OMIM BF4 has so far been used as a solvent for the extraction of thiophene and its derivatives from gasoline [76], as an ecological medium for organic synthesis [77,78], and as an extractant for the extraction of phenolic compounds [79].

The proposed extraction method utilizing OMIM BF4 allows the removal of up to 94.89% of heme and/or hemoglobin from the serum samples. The serum after heme/hemoglobin removal by this method does not change the composition of the remaining blood proteins and allows to improve the precision and accuracy of the ELISA test for cotinine content to over 30%. The mechanism of removal was interpreted in terms of a series of molecular modeling calculations and was found to rely on the enhanced affinity of heme/hemoglobin to OMIM BF4 in comparison to water and the opposite relation in the case of cotinine.

Our method offers many relevant advantages of sample preparation for the cotinine test by removing heme/hemoglobin, for instance, high extraction efficiency, selectivity, mild operating conditions, short equilibration time, and a biocompatible environment. In comparison to other methods dedicated to the selective removal of trace hemoglobin, which are described in the introduction part, our method is a very fast-extraction method and requires a short equilibration time of 5 min. It is three times smaller in comparison to MNPs [52]. Secondly, it utilized commercially available IL, so the multi-stage synthesis of tailored IL or MNPs can be omitted. Thirdly, obtained recovery of hemoglobin (~95%) is slightly better than that declared by Cheng et al. (~93%) utilizing correctly designed IL for target applications [55]. Based on the cotinine test results, it can be anticipated that the selective removal of heme/hemoglobin from human blood is able to minimize interference as well in other diagnostic assays.

## 5. Patents

The Polish Patent P.442168 was resulted from the work reported in this manuscript.

## Figures and Tables

**Figure 1 ijms-23-13692-f001:**
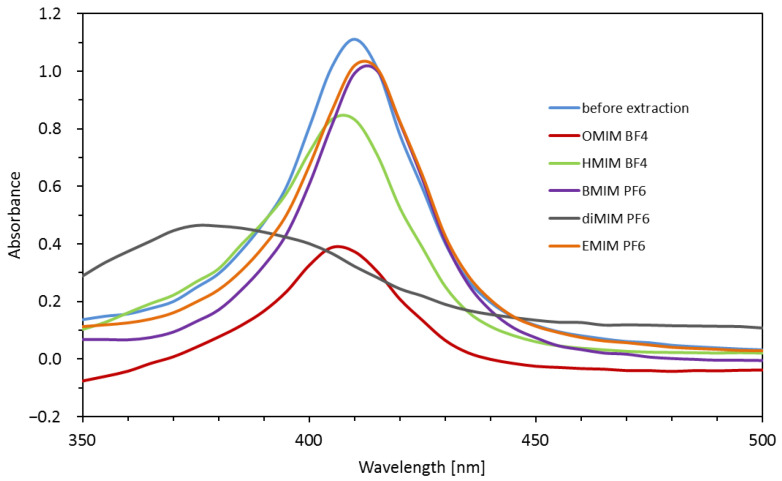
UV–vis spectra recorded in the range of 350–500 nm for the serum samples: before extraction; and after extraction with ILs.

**Figure 2 ijms-23-13692-f002:**
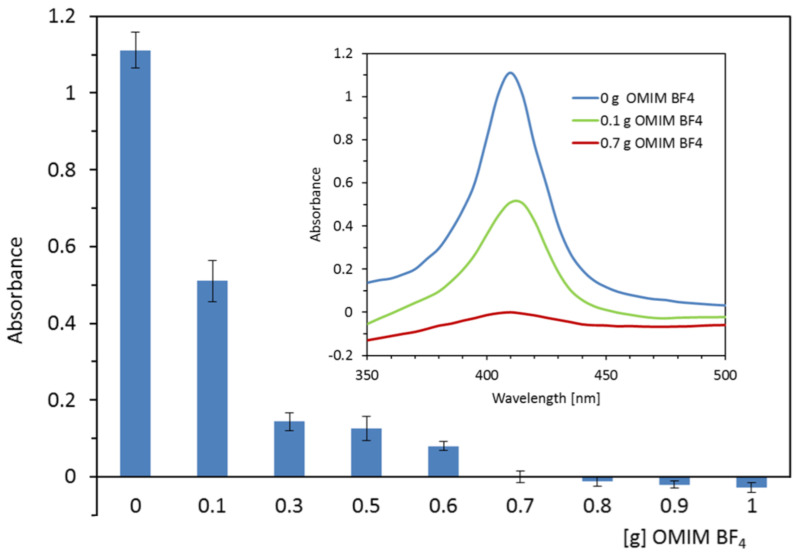
The dependence of the absorbance at 410 nm of the serum sample containing hemoglobin on the mass of the OMIM BF_4_ used for the extraction. Insert represents the spectrum of the serum before extraction (0 g OMIM BF_4_) and after extraction with OMIM BF_4_.

**Figure 3 ijms-23-13692-f003:**
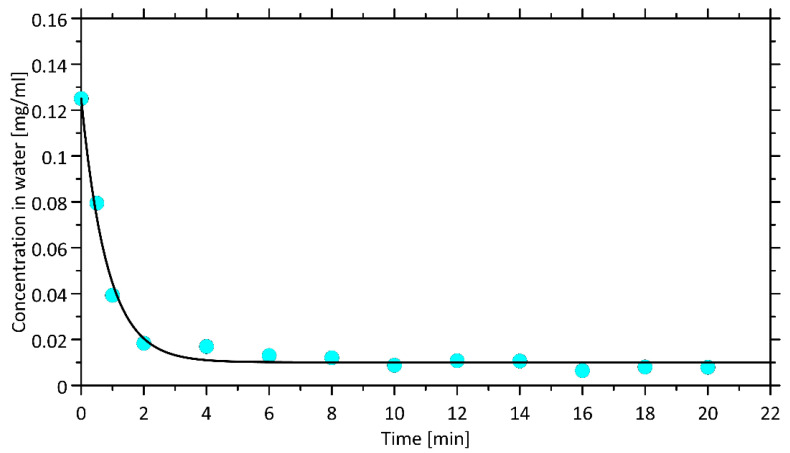
Dependence of the shaking time of the extraction mixture on the concentration of hemoglobin in the aqueous phase. The solid line corresponds to the predictions of the integral form of Equation (1).

**Figure 4 ijms-23-13692-f004:**
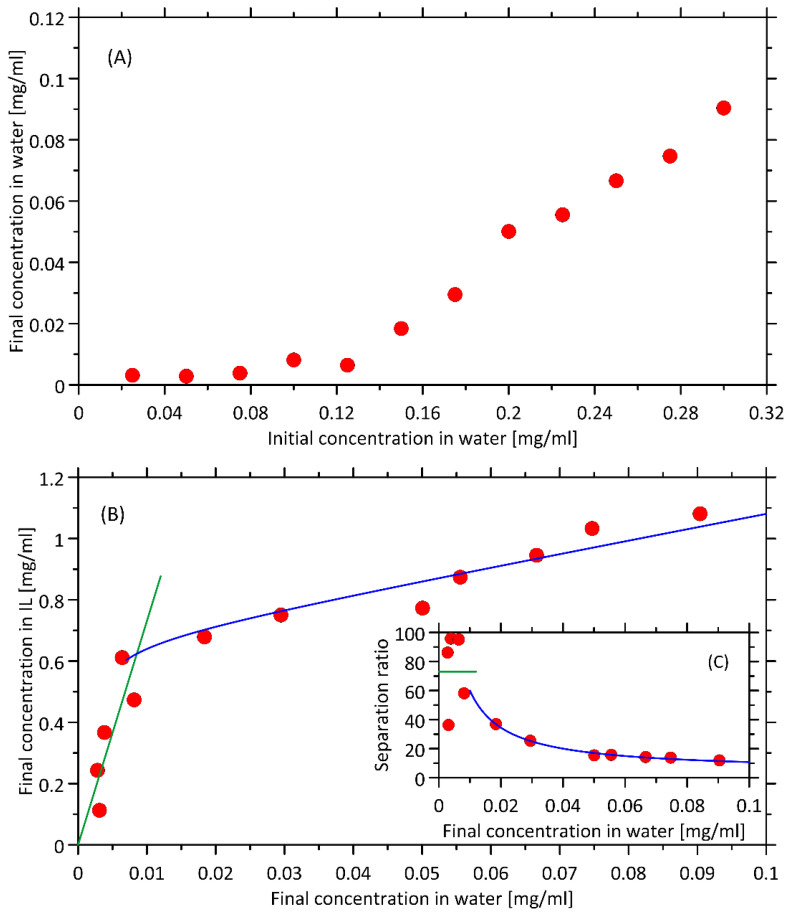
(**A**) Dependence of the hemoglobin concentration measured after extraction with the OMIM BF_4_ on the initial concentration of hemoglobin in a blood serum sample. (**B**) The steady state partitioning behavior of hemoglobin between the two phases. The y-axis is the concentration of hemoglobin in the bottom phase, and the x-axis is the concentration of the hemoglobin in the upper phase. The green and blue solid lines denote the predictions of the models assuming either ideal, proportional distribution of heme between two phases or the partial saturation of the ionic liquid phase by dissolved heme, respectively. (**C**) The separation ratio calculated as the ratio of final concentrations of heme in both solvents.

**Figure 5 ijms-23-13692-f005:**
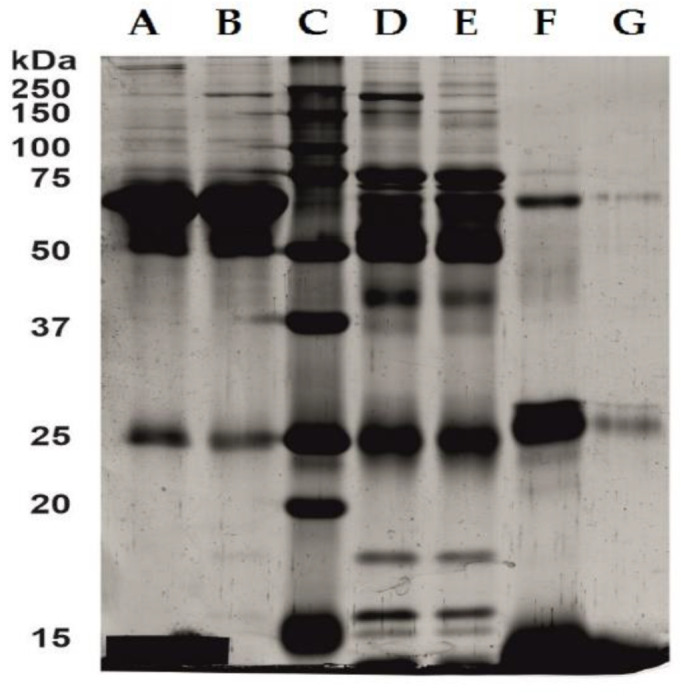
Electrophoretic profiles (SDS-PAGE): A—serum enriched with hemoglobin before extraction, B—serum enriched with hemoglobin after extraction with the OMIM BF_4_, C—standard of molecular weights, D—serum enriched with hemoglobin after removal of albumin before extraction, E—serum enriched with hemoglobin after removal of albumin after extraction with the OMIM BF_4_, F—20× concentrated serum enriched with hemoglobin before extraction, G—20× concentrated serum enriched with hemoglobin after extraction with the OMIM BF_4_.

**Figure 6 ijms-23-13692-f006:**
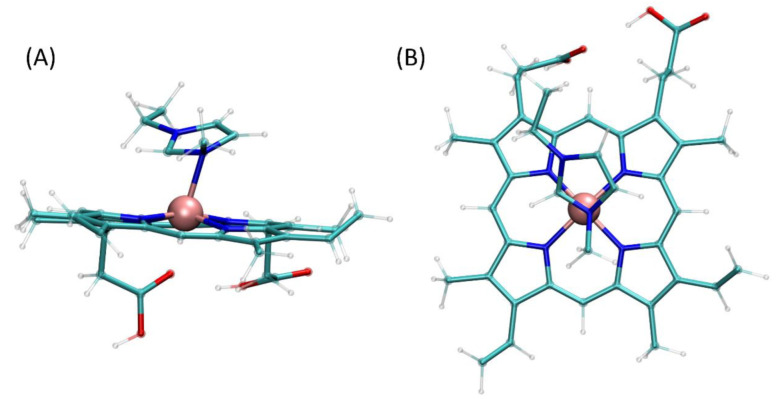
The structure of the heme–OMIM complex, identified and optimized at the B3LYP/6-311G(d,p) level of theory. View along two different axes (**A**,**B**).

**Figure 7 ijms-23-13692-f007:**
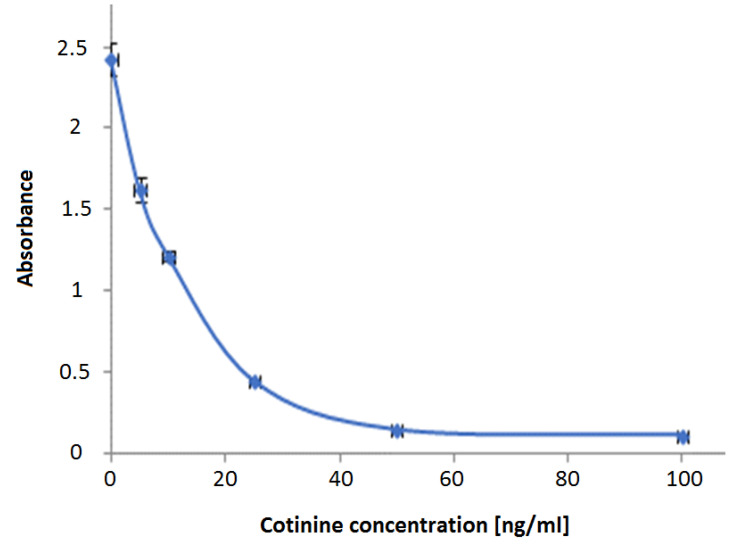
Calibration curve illustrating the dependence of absorbance vs. cotinine concentration [ng mL^−1^].

**Figure 8 ijms-23-13692-f008:**
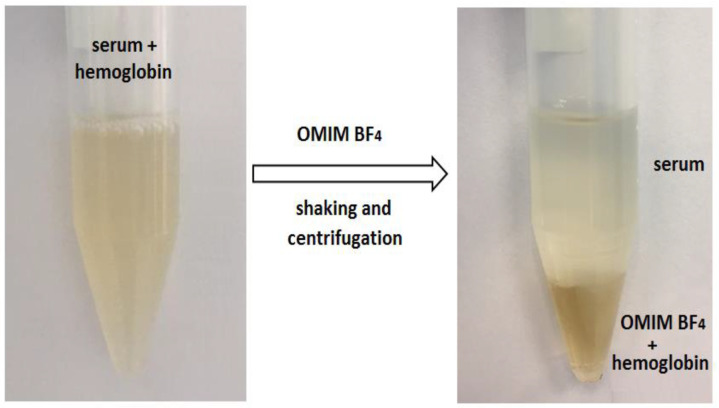
Biphasic system created after mixing 0.7 g OMIM BF_4_ and 3 mL serum enriched with 0.15 mg mL^−1^ of hemoglobin.

**Table 1 ijms-23-13692-t001:** Results of measurements of cotinine content in serum samples containing hemoglobin at three concentration levels (0.05 mg mL^−1^; 0.1 mg mL^−1^; 0.2 mg mL^−1^) and selected cotinine concentration (20 ng mL^−1^) without and with extraction of heme/hemoglobin using the Cotinine ELISA Kit test.

Mode of Measurements	Parameter	Hemoglobin Concentration in Serum Samples			
		0 mg mL^−1^	0.05 mg mL^−1^	0.1 mg mL^−1^	0.2 mg mL^−1^
Measurements with the cotinine ELISA Kit without removing heme/hemoglobin	Absorbance	0.562 0.652 0.517	0.560 0.519 0.445	0.465 0.521 0.580	0.345 0.442 0.396
	Concentration [ng mL^−1^]	21.904 19.950 22.971	21.955 22.917 24.867	24.314 22.891 21.496	28.042 24.953 26.314
	Concentration mean value [ng mL^−1^]	21.608	23.246	22.900	26.436
	SD	1.532	1.484	1.409	1.548
	CV%	7.090	6.382	6.153	5.856
	Relative error [%]	8.04	16.23	14.5	32.18
Measurements with the cotinine ELISA Kit after heme/hemoglobin extraction	Absorbance	-	0.625 0.546 0.493	0.830 0.803 0.759	0.625 0.642 0.684
	Concentration [ng mL^−1^]	-	20.523 22.288 23.588	16.464 16.964 17.793	20.515 20.145 19.287
	Concentration mean value [ng mL^−1^]	-	22.133	17.074	19.982
	SD	-	1.538	0.671	0.630
	CV%	-	6.951	3.932	3.153
	Relative error [%]	-	10.66	14.6	−0.09

## Data Availability

Not applicable.

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
