# Peer review of "Influence of Selective Extraction/Isolation of Heme/Hemoglobin with Hydrophobic Imidazolium Ionic Liquids on the Precision and Accuracy of Cotinine ELISA Test"

_ijms, 2022, doi:10.3390/ijms232213692_

Round 1

Reviewer 1 Report

In this article, the authors present the obtention of good-quality serum samples for the determination of cotinine concentration using the ELISA test by removing trace amounts of heme and/or hemoglobin, by the use imidazolium ionic. Also, they tried quantum mechanical calculations, classical molecular dynamics simulations, and thermodynamic integration calculations to provide a molecular interpretation of processes.

Consequently, the topic is suitable for publication and interesting for the development of cheaper methods for extraction of hemoglobin from blood in order to determine cotinine concentration by ELISA technique.

Therefore, I recommend a minor revision, which is mainly based on the following weaknesses of the article:

In Figure 1: It would be better if the correspondence between each color and the solvent were inside the scheme and not in the legend.

I consider that the title of every figure and tables are too long. They have methodological information that can be said in the prior pharagraph.

The authors must discuss the findings of the results comparing with recent literature in the introduction section to strengthen the work, thus I consider this section is poor and it is only the presenting of the results with no comparation with previous works.

Please, enhance the discussion regarding the not fitting of the kinetic and equilibrium data to the first-order reversible model, as well as the not description of the data in the range of evaluated concentrations. Is there another model that can be implemented? What software did you use for modeling?

The conclusion must be reviewed and adjusted, considering the discussion of the results respect previous works published.

Author Response

In this article, the authors present the obtention of good-quality serum samples for the determination of cotinine concentration using the ELISA test by removing trace amounts of heme and/or hemoglobin, by the use imidazolium ionic. Also, they tried quantum mechanical calculations, classical molecular dynamics simulations, and thermodynamic integration calculations to provide a molecular interpretation of processes.

Consequently, the topic is suitable for publication and interesting for the development of cheaper methods for extraction of hemoglobin from blood in order to determine cotinine concentration by ELISA technique.

Therefore, I recommend a minor revision, which is mainly based on the following weaknesses of the article:

In Figure 1: It would be better if the correspondence between each color and the solvent were inside the scheme and not in the legend.

Answer: Thank You for this suggestion. We corrected Figure 1 accordingly.

I consider that the title of every figure and tables are too long. They have methodological information that can be said in the prior paragraph.

Answer: Thank You for this suggestion. We have shorted the Figures and Tables' legends.

The authors must discuss the findings of the results comparing with recent literature in the introduction section to strengthen the work, thus I consider this section is poor and it is only the presenting of the results with no comparation with previous works.

Answer: Thank You for this suggestion. We added new references coming from the latest publications to rise the possibility of relating obtained results with those that already exist. The improved section:

To separate histidine-rich proteins, such as hemoglobin, affinity chromatography on transition metal ions (Cu2+, Ni2+, Zn2+ or Co2+) immobilized on matrices can be applied  [48]. Metals could also be immobilized on the surface of magnetic nanoparticles (MNP) [49]. Magnetic cores, usually made of iron oxides, are coated with SiO2 for protection against acidic environments [50]. An example of such a composite is eg CoFe2O4, [51]. More recently hierarchical copper shells anchored on magnetic nanoparticles were designed to selectively deplete hemoglobin from human blood [52]. The batch experiments were conducted using bovine hemoglobin (BHb). The adsorption increased linearly with increasing initial protein concentration up to 1.0 mg mL−1. The adsorption capacity was estimated to be 666 mg BHb per 1 g of Cu-MNPs. The system required however long equilibration time of about 90 min and 100-fold sample dilution in 0.1 M Tris-HCl pH 8.5.

Another possibility of the purification and separation of proteins offers extraction systems based on ionic liquids such as liquid-liquid extraction utilizing hydrophobic ionic liquids, two-phase aqueous systems (ABS) consisting of hydrophilic ionic liquids, and supported ionic liquids (SILs), in which ILs are modifying different solid materials as covalently bonded ligands. Excellent review articles [53,54] have been provided on this subject. In 2008, Cheng et al. [55] for the first time extracted hemoglobin from serum samples by a new synthesized ionic liquid 1-butyl-3-trimethylsilyl-imidazolium hexafluorophosphate (BtmsimPF6). The authors managed to remove 100 ng µL-1 of hemoglobin with an efficiency of 93%. However, the preparation of BtmsimPF6 required a tedious, time-consuming,  procedure with the use of different organic solvents such as chlorobutane, toluene, ethyl acetate, and controlling the temperature. The yield at last stage of the multi-stage synthesis process namely the conversion of chloride to hexafluorophosphate ionic liquid achieved 85%.

New References:

  1. Wang, S.; Xiong, N.; Dong, X.Y.; Sun, Y. A novel nickel-chelated surfactant for affinity-based aqueous two-phase micellar extraction of histidine-rich protein. Chromatogr. A 2013, 1320, 118-24. doi: 10.1016/j.chroma.2013.10.074.
  2. Zhang, M.; He, X.; Chen, L.; Zhang, Y. Preparation and characterization of iminodiacetic acid-functionalized magnetic nanoparticles and its selective removal of bovine hemoglobin. Nanotechnology 2011, 22, 065705. doi: 10.1088/0957-4484/22/6/065705.
  3. Guo, X.; Mao, F.; Wang, W.; Yang, Y.; Bai, Z. Sulfhydryl-modified Fe3O4@SiO2 core/shell nanocomposite: synthesis and toxicity assessment in vitro. ACS Appl. Mater. Interfaces 2015, 7, 14983–14991. doi: 10.1021/acsami.5b03873.
  4. Ayyappan, S.; Mahadevan, S.; Chandramohan, P.; Srinivasan, M. P.; Philip, J.; Raj, B. Influence of Co2+ Ion concentration on the size, magnetic properties, and purity of CoFe2O4 Spinel ferrite nanoparticles. Phys. Chem. C 2010, 114, 6334–6341. doi: 10.1021/jp911966p.
  5. Liu, Y.; Wang, Y.; Yan, M.; Huang, J. Selective Removal of Hemoglobin from Blood Using Hierarchical Copper Shells Anchored to Magnetic Nanoparticles. Biomed Res. Int. 2017, 2017, Article ID 7309481. doi: 10.1155/2017/7309481.
  6. Schröder, C. Proteins in Ionic Liquids: Current Status of Experiments and Simulations. Curr. Chem. (Cham) 2017, 375, 25. doi: 10.1007/s41061-017-0110-2.
  7. Bento, R.M.F.; Almeida, C.A.S.; Neves, M.C.; Tavares, A.P.M.; Freire, M.G. Advances Achieved by Ionic-Liquid-Based Materials as Alternative Supports and Purification Platforms for Proteins and Enzymes. Nanomaterials (Basel) 2021, 11, 2542. doi: 10.3390/nano11102542.

Please, enhance the discussion regarding the not fitting of the kinetic and equilibrium data to the first-order reversible model, as well as the not description of the data in the range of evaluated concentrations. Is there another model that can be implemented? What software did you use for modeling?

Answer: Actually, both the kinetic and equilibrium data *are* fitted which is graphically illustrated in Figs. 3 and 4, respectively. More precisely, the kinetic data were fitted by using the integral form of eq. (1) whereas the equilibrium data were fitted in two separate ranges by using the two different mathematical formulae (constant ratio of concentrations in two phases for low concentration and empirical, hyperbolic relationship between the separation ration and solute concentration for larger concentrations). In spite of several attempts we were unable to propose the equilibrium-related model that: (i) would have a well-defined physical meaning; (ii) would provide a set of best-fitted parameters of realistic values; (iii) would work well for the whole range of concentrations. From the course of the ‘concentration in water vs. concentration in IL’ curve we deduced that the observed, non-typical behavior is the results of some saturation processes within the IL phase, which is indicated in the current version of the paper. Regarding the last question, we used the Mathematica 12 (Wolfram Res.) for mathematical modeling (including both fitting and solving the differential equation (1)). This was included in the revised version of the paper.

The conclusion must be reviewed and adjusted, considering the discussion of the results respect previous works published.

Answer: We mentioned in the conclusion the achievements described in the previous papers. We added the following part of conclusion:

Our method offers many relevant advantages of sample preparation for cotinine test by removing heme/hemoglobin: for instance, high extraction efficiency, selectivity, mild operating conditions, short equilibration time, and a biocompatible environment. In comparison to other methods dedicated to the selective removal of trace hemoglobin, which are described in the introduction part, our method is very fast-extraction and requires a short equilibration time 5 min. It is three times smaller in comparison to MNPs [52]. Secondly, it utilized commercially available IL, so the multi-stage synthesis of tailored IL or MNPs can be omitted. Thirdly, obtained recovery of hemoglobin (~95%) is a little bit better than that one declared by Cheng et al. (~93%) utilizing correctly designed IL for target applications [55]. Based on the cotinine test results it can be anticipated that the selective removal of heme/hemoglobin from human blood is able to minimize interference as well in other diagnostic assays.

Reviewer 2 Report

The authors of the manuscript "Influence of selective extraction/isolation of heme/hemoglobin with hydrophobic imidazolium ionic liquids on the precision and accuracy of Cotinine ELISA test" presents a very interesting study. The article is written in a clear and understandable way. The selection of literature is appropriate and logical, there are no unnecessary citations.

However, the article has major flaws in its current form.

1.       What is the novelty of this research article?  In the introduction section, the author states “In 2008, Cheng et al. [48] for the first time extracted hemoglobin from serum samples by a new synthesized ionic liquid 1-butyl-3-trimethylsilyl-imidazolium hexafluorophosphate (BtmsimPF6). The authors managed to remove 100 ng μL-1 of hemoglobin with an efficiency of 93%.” It has a comparable result with the author’s finding.

2.       On pages 3-4, the author did not explain why 0.15 mg/mL or 0.125 mg/mL of hemoglobin is chosen for optimization of extraction conditions.  According to the UCSF Health website, plasma or serum in someone who does not have hemolytic anemia may contain up to 5 milligrams per deciliter (mg/dL) or 0.05 mg/mL.  With lower concentrations, the extraction efficiency may become worse. 

3.       Figure 4: how many replications for each concentration?

4.       On page 9, the author states “ The concentration of 20 ng mL-1, lying in the middle of the curve, was chosen to investigate the effect of heme/hemoglobin on the accuracy and precision of the serum cotinine determination. This is a critical value used to distinguish smokers from non-smokers.”  Any references for this statement?  According to the CDC website, cotinine can be measured in serum, urine, saliva, and hair. Nonsmokers exposed to typical levels of secondhand tobacco smoke (SHS)  have serum cotinine levels of less than 1 ng/mL, with heavy exposure to SHS producing levels in the 1–10 ng/mL range. Active smokers almost always have levels higher than 10 ng/mL and sometimes higher than 500 ng/mL (Hukkanen et al., 2005).

5.       Table 1 data and figure 4 results have contradictory results.  According to figure 4, with 0.2 mg/mL initial hemoglobin concentration, the final concentration is around 0.05 mg/mL.  In theory, the data for hemoglobin concentration (0.2 mg/mL after heme/hemo extraction) in serum samples should have comparable results with 0.05 mg/mL without extraction samples.  Has the author considered more replications?  With random results, it’s very challenging to draw persuasive conclusions.

6.       Is this any explanation for why the 0.2 mg/mL samples have better results than 0.05 or 0.1mg/mL samples after the extraction?

7.       Is this any explanation for why no improvement is achieved for 0.1 mg/mL samples (with and without extractions) 

Author Response

The authors of the manuscript "Influence of selective extraction/isolation of heme/hemoglobin with hydrophobic imidazolium ionic liquids on the precision and accuracy of Cotinine ELISA test" presents a very interesting study. The article is written in a clear and understandable way. The selection of literature is appropriate and logical, there are no unnecessary citations.

However, the article has major flaws in its current form.

  1. What is the novelty of this research article? In the introduction section, the author states “In 2008, Cheng et al. [48] for the first time extracted hemoglobin from serum samples by a new synthesized ionic liquid 1-butyl-3-trimethylsilyl-imidazolium hexafluorophosphate (BtmsimPF6). The authors managed to remove 100 ng μL-1 of hemoglobin with an efficiency of 93%.” It has a comparable result with the author’s finding.

Answer: 1.          Cheng et al. used the newly synthesized ionic liquid 1-butyl-3-trimethylsilyl-imidazolium hexafluorophosphate (BtmsimPF6) for the extraction. This substance is not commercially available, which limits the possibility of wider application of the method described by Cheng. The synthesis procedure is as follows:

„1-Butyl-3-trimethylsilylimidazolium hexafluorophosphate (BtmsimPF6) was prepared by adopting a documented pathway with minor modifications [Z.J. Li, Q. Wei, R. Yuan, X. Zhou, H.Z. Liu, H.X. Shan, Q.J. Song Talanta, 71 (2007), p. 68]. 1-Butyl-3-trimethylsilylimidazolium chloride (BtmsimCl) was first prepared by the reaction of 0.5 mol 1-trimethylsilylimidazole and 0.5 mol chlorobutane in 50 mL of toluene in a nitrogen atmosphere at 90 °C for 72 h with stirring and refluxing. The obtained transparent viscous BtmsimCl was cooled and then washed with 50 mL of ethyl acetate for three times, followed by drying at 80 °C. 0.5 mol of BtmsimCl was mixed with 100 mL of water in a flask, to which 100 mL of hexafluorophosphoric acid (63%) was added drop-wise in order to control the temperature not to exceed 50 °C. After the mixture was stirred vigorously for 1 h, the ionic liquid phase in the bottom of the flask was separated and washed with a sufficient amount of water until a neutral wash out solution was obtained, i.e., pH 6.5. The BtmsimPF6 was finally dried at 80 °C under vacuum for 24 h. The yield at this stage, i.e., the conversion of chloride to hexafluorophosphate ionic liquid, was 85%”.

In our work, for the first time in the extraction of heme/hemoglobin, a commercially available ionic liquid (1-methyl-3-octylimidazolium tetrafluoroborate) was used. The use of a commercially available ionic liquid makes the described method cheaper, simpler and accessible to most laboratories.

  1. On pages 3-4, the author did not explain why 0.15 mg/mL or 0.125 mg/mL of hemoglobin is chosen for optimization of extraction conditions. According to the UCSF Health website, plasma or serum in someone who does not have hemolytic anemia may contain up to 5 milligrams per deciliter (mg/dL) or 0.05 mg/mL.  With lower concentrations, the extraction efficiency may become worse.

Answer: The process of hemolysis may take place in vivo as a result of autoimmune hemolytic anemia, as a post-transfusion reaction, and in vitro as a result of improper collection, handling, or transport of samples. Therefore, the hemoglobin concentration in the samples may reach values higher than 0.05 mg / lm, and such concentrations have been used in the performed experiments. In addition, the efficiency of the extraction process typically deteriorates as the concentration of the analyte in the sample increases. The use of higher than expected hemooglobin concentrations made it possible to rule out this.

  1. Figure 4: how many replications for each concentration?

Answer: Experiments were performed in triplicate for each concentration.

  1. On page 9, the author states “ The concentration of 20 ng mL-1, lying in the middle of the curve, was chosen to investigate the effect of heme/hemoglobin on the accuracy and precision of the serum cotinine determination. This is a critical value used to distinguish smokers from non-smokers.” Any references for this statement?  According to the CDC website, cotinine can be measured in serum, urine, saliva, and hair. Nonsmokers exposed to typical levels of secondhand tobacco smoke (SHS)  have serum cotinine levels of less than 1 ng/mL, with heavy exposure to SHS producing levels in the 1–10 ng/mL range. Active smokers almost always have levels higher than 10 ng/mL and sometimes higher than 500 ng/mL (Hukkanen et al., 2005).

Answer: "Cotinine ELISA Kit" has a sensitivity of 1 ng mL-1 so we can not measure lower cotinine contents.Higher contents require dilution of samples to achieve a concentration located within the calibration curve range. Answering the last question, We mentioned in the introduction the Kim' work [Kim, S. Overview of Cotinine Cutoff Values for Smoking Status Classification. Int J Environ Res Public Health 2016, 13, 1236. doi:10.3390/ijerph13121236] where 10-20 ng/ml of cotinine in serum has been proposed as a critical value used to distinguish smokers from non-smokers. However, we agree with the reviewer that determining the limit value is an open matter. We chose a value based on extensive long-term epidemiological studies.

  1. Table 1 data and figure 4 results have contradictory results. According to figure 4, with 0.2 mg/mL initial hemoglobin concentration, the final concentration is around 0.05 mg/mL.  In theory, the data for hemoglobin concentration (0.2 mg/mL after heme/hemo extraction) in serum samples should have comparable results with 0.05 mg/mL without extraction samples.  Has the author considered more replications?  With random results, it’s very challenging to draw persuasive conclusions.

Answer:5.           The concentration of 0.05 mg/ml in Table 1 is also the starting concentration. After the extraction process, the hemoglobin concentration in the sample also changed in this case. Therefore, the results obtained for samples with a starting hemoglobin concentration of 0.2 mg/ml and 0.05 mg / ml are different.

  1. Is this any explanation for why the 0.2 mg/mL samples have better results than 0.05 or 0.1mg/mL samples after the extraction?

Answer:6.           Probably the best results for the samples with the highest hemoglobin content are associated with its negative influence on the test results. The more interfering hemoglobin in the sample, the worse the results of the cotinine determination. Removal of 0.2 mg/ml of hemoglobin from the sample improves the cotinine results the most.

  1. Is this any explanation for why no improvement is achieved for 0.1 mg/mL samples (with and without extractions)

Answer: We have shown the results we obtained from the conducted experiments. It is known that the relative measurement error is a certain compensation, or rather the resultant of the errors in + and in -. However, I admit that this is an interesting question and seems to be related to the shape of the calibration curve.

Round 2

Reviewer 2 Report

  1. Table 1 data and figure 4 results have contradictory results. According to figure 4, with 0.2 mg/mL initial hemoglobin concentration, the final concentration is around 0.05 mg/mL.  In theory, the data for hemoglobin concentration (0.2 mg/mL after heme/hemo extraction) in serum samples should have comparable results with 0.05 mg/mL without extraction samples.  Has the author considered more replications?  With random results, it’s very challenging to draw persuasive conclusions.

Answer:5.           The concentration of 0.05 mg/ml in Table 1 is also the starting concentration. After the extraction process, the hemoglobin concentration in the sample also changed in this case. Therefore, the results obtained for samples with a starting hemoglobin concentration of 0.2 mg/ml and 0.05 mg / ml are different.

How about the 0.05 mg/mL WITHOUT extraction, the concentration of hemoglobin should be 0.05 mg/mL which should get comparable results with 0.2 mg/mL sample that has gone through extraction. 

Has the author considered the extraction impact on Nicotine detection?  In table 1, the author did not show the results for 0 mg/mL sample after the extraction process.  Will the extraction process lower the nicotine concentration? 

Author Response

  1. How about the 0.05 mg/mL WITHOUT extraction, the concentration of hemoglobin should be 0.05 mg/mL which should get comparable results with a 0.2 mg/mL sample that has gone through extraction.

If the initial concentration is 0.2 mg/ml, the concentration of hemoglobin after extraction should be certainly about 0.05 mg/ml - this can be seen in Fig. 4. Therefore, we absolutely agree with the reviewer's deduction. It should be remembered, however, that Fig. 4 is related to the changes in hemoglobin concentration but table 1 shows the results for the cotinine determination. The question is - If the anticipated hemoglobin concentration has been finally comparable for samples with and without extraction, would the amount of cotinine have been the same?  Looking at the results - they're not the same. The results obtained, in the columns indicated by the reviewer, differ by about 16%.  In our opinion, the differences are due to the fact that the samples were prepared differently. The extraction required extra steps and adding a new reagent (ionic liquid).

The Cotinine Elisa kit belonging to Immunoassays appeared to be sensitive to the presence of a trace amount of heme/hemoglobin. It was the aim of this paper. To prove the impact of heme/hemoglobin on the test accuracy, as a blank, in both cases, we used cotinine-enriched serum samples without hemoglobin and therefore without needed extraction.

  1. Has the author considered the extraction impact on Nicotine detection? In table 1, the author did not show the results for 0 mg / mL sample after the extraction process. Will the extraction process lower the nicotine concentration?

We did not perform this test as explained above. We assumed that we could make one blank for both samples. However, the reviewer's remark is inspiring. It made us realize the need for further research to study the effect of traces of various ionic liquids on the results of immunoassay.